# NlpC/P60 peptidoglycan hydrolases of *Trichomonas vaginalis* have complementary activities that empower the protozoan to control host-protective lactobacilli

**Michael J. Barnett[1], Jully Pinheiro[1], Jeremy R. Keown[1], Jacob Biboy[2], Joe Gray[3], Ioana-Wilhelmina Lucinescu[3], Waldemar Vollmer[2], Robert P. Hirt[3], Augusto Simoes-Barbosa[1]☯\*, David C. Goldstone[1,4]☯\***

1 School of Biological Sciences, University of Auckland, Auckland, New Zealand, 2 Centre for Bacterial Cell Biology, Biosciences Institute, Newcastle University, Newcastle upon Tyne, United Kingdom, 3 Biosciences Institute, Newcastle University, Newcastle upon Tyne, United Kingdom, 4 Maurice Wilkins Centre for Molecular Biodiscovery, Auckland, New Zealand

☯ These authors contributed equally to this work.
\* a.barbosa@auckland.ac.nz (AS-B); d.goldstone@auckland.ac.nz (DCG)

**Data Availability Statement:** All relevant data are within the manuscript and Supporting Information files.

## Abstract

*Trichomonas vaginalis* is a human protozoan parasite that causes trichomoniasis, a prevalent sexually transmitted infection. Trichomoniasis is accompanied by a shift to a dysbiotic vaginal microbiome that is depleted of lactobacilli. Studies on co-cultures have shown that vaginal bacteria in eubiosis (e.g. *Lactobacillus gasseri*) have antagonistic effects on *T. vaginalis* pathogenesis, suggesting that the parasite might benefit from shaping the microbiome to dysbiosis (e.g. *Gardnerella vaginalis* among other anaerobes). We have recently shown that *T. vaginalis* has acquired NlpC/P60 genes from bacteria, expanding them to a repertoire of nine TvNlpC genes in two distinct clans, and that TvNlpCs of clan A are active against bacterial peptidoglycan. Here, we expand this characterization to TvNlpCs of clan B. In this study, we show that the clan organisation of NlpC/P60 genes is a feature of other species of *Trichomonas*, and that *Histomonas meleagridis* has sequences related to one clan. We characterized the 3D structure of TvNlpC_B3 alone and with the inhibitor E64 bound, probing the active site of these enzymes for the first time. Lastly, we demonstrated that TvNlpC_B3 and TvNlpC_B5 have complementary activities with the previously described TvNlpCs of clan A and that exogenous expression of these enzymes empower this mucosal parasite to take over populations of vaginal lactobacilli in mixed cultures. TvNlpC_B3 helps control populations of *L. gasseri*, but not of *G. vaginalis*, which action is partially inhibited by E64. This study is one of the first to show how enzymes produced by a mucosal protozoan parasite may contribute to a shift on the status of a microbiome, helping explain the link between trichomoniasis and vaginal dysbiosis. Further understanding of this process might have significant implications for treatments in the future.

**Funding:** This work was supported by the Maurice & Phyllis Paykel Trust and the Faculty Research Development Fund from the University of Auckland (AS-B). WV was supported by the BBSRC (BB/W013630/1). The funders had no role in study design, data collection and analysis, decision to publish, or preparation of the manuscript.

**Competing interests:** The authors have no competing interest.

## Author summary

Trichomoniasis is a frequent infection of the human urogenital tract. The causative agent is *Trichomonas vaginalis*, a protozoan parasite that lives on the mucosal surface of the vagina and has to interact with bacteria naturally residing on this site (i.e. microbiota). It has been shown that the composition of bacterial species of the vaginal microbiota differs between infected and non-infected women, but what drives this microbial shift during infection to an unbalanced microbiota (i.e. dysbiosis) remains unknown. A previous study from our group revealed that *T. vaginalis* acquired NlpC/P60 genes from bacteria (or TvNpCs), known for producing enzymes that break down peptidoglycan which is the main component of bacterial cell walls. Here, we expand this characterization to show that different TvNpCs have complementary activities helping the protozoan to control populations of vaginal lactobacilli in mixed cultures in laboratory. This study advances our knowledge on the interaction of mucosal pathogens with the microbiota, a topic of increasingly importance for improving treatments including against trichomoniasis and morbidities associated with a dysbiotic microbiota.

## Introduction

*Trichomonas vaginalis* is an extracellular protozoan parasite that infects the urogenital tract of humans causing trichomoniasis. With about a quarter of a billion cases each year, trichomoniasis ranks as the most common nonviral sexually transmitted infection in the world affecting more people than Syphilis, *Neisseria gonorrhoeae* and *Chlamydia trachomatis* combined [1]. Trichomoniasis causes vaginitis, prostatitis and urethritis with women carrying a disproportional morbidity burden including increased chances of developing pelvic inflammatory disease [2], cervical cancer [3, 4], increased risk of acquiring and transmitting HIV [5, 6] and, concerningly, significantly adverse pregnancy outcomes [2–4, 7, 8].

The vaginal microbiome of women of reproductive age has been classified into five community state types (CSTs), where the dominance of a single species of *Lactobacillus* has been allocated to four of these communities and representing ~75% of asymptomatic women [9]. Trichomoniasis has been correlated with the species-diversified CST where, conversely, host-protective lactobacilli are excluded or less dominant [10, 11]. This CST is formed by mostly anaerobic bacteria with the dominance of *Prevotella*, *Gardnerella*, *Megasphera*, *Snethia* and *Atopobium vaginae* coincidently found in women experiencing a dysbiotic condition known as bacterial vaginosis [9–11]. These observations do not necessarily imply on causation, but at least suggest that *T. vaginalis* may shape this microbiome [12]. Indeed, vaginal lactobacilli exert protective effects against *T. vaginalis* [13, 14] while vaginal dysbiotic bacteria engage synergistically with this protozoan [15–17]. Whether a causation relationship may exist between *T. vaginalis* and the status of the vaginal microbiome, the underlying molecular mechanisms of this microbial interaction remain poorly understood.

The adaptation of *T. vaginalis* to the urogenital tract of humans is thought to be recent, relative to evolution of trichomonads as parasites of various hosts and mucosal sites, and was followed by genome expansion including the lateral acquisition of genes from bacteria [18–20]. We have recently identified that *T. vaginalis* acquired bacterial NlpC/P60 enzymes, likely from two events of lateral gene transfer and followed by gene duplication resulting in nine genes that fall into clans A and B [21]. NlpC/P60 proteins were identified early on as cysteine proteases (clan CA, family C40) [18], and broadly classified under the cysteine/histidine dependant aminohydrolase/peptidase super family (CHAP) [22]. NlpC/P60-containing enzymes typically

hydrolyse the stem peptides of bacterial peptidoglycan (PG). The expansion of this gene family in *T. vaginalis*, a eukaryotic organism that lacks PG, suggests these enzymes may play important role in mediating interactions between the parasite and the bacterial members of the human urogenital tract microbiota.

The lateral acquisition of bacterial genes targeting PG has been observed in other eukaryotes [23], and considered as an adaptive trait to target specific bacteria [24]. Therefore, targeting PG might represent one of the mechanisms by which *T. vaginalis* shapes the vaginal microbiota contributing to the parasite's capacity to thrive at this mucosal site. It is also possible this trait may have even evolved earlier in this lineage, and thus shared across other species of trichomonads that are known for parasitizing other mucosal sites and hosts. Although this has yet to be demonstrated, infection by *Trichomonas gallinae* that parasitizes the mucosa of the upper digestive tract of pigeons was found to cause significant alterations on the gut microbiota with depletion of lactobacilli [25].

We have previously shown that two members of *T. vaginalis* NlpC/P60 enzymes, both belonging to clan A, are functional DL-endopeptidases degrading stem peptides of PG [21]. These enzymes, TvNlpC_A1 and TvNlpC_A2, are secreted by the protozoan and may confer an adaptive advantage for controlling and/or exploiting bacterial populations. Here, we extend the characterization of TvNlpCs to two other members of the clan B (TvNlpC_B3 and B5). We show the presence of the two gene family clans in other trichomonads, characterize the structure of one the *T. vaginalis* clan B enzyme with and without a ligand, and we demonstrate that TvNlpCs of clans A and B have complementary cellular and enzymatic capabilities that support *T. vaginalis* on controlling populations of vaginal lactobacilli in mixed cultures.

## Results

### The diversity and phylogeny of NlpC/P60 genes in *T. vaginalis* and other trichomonads

A total of nine genes encoding NlpC/P60 homologues were initially annotated in the draft genome of *Trichomonas vaginalis* [18] and detailed sequence comparisons identified two distinct subfamilies, called clans A and B, with respective members named TvNlpC_A1-4 (four members) and TvNlpC_B1-5 (five members) (S1A Table), derived from likely two distinct lateral gene transfers from distinct Gram-positive bacteria donors [21, 26]. Published transcriptomics data [27, 28] indicated that at least one member of each family is expressed in different strains, with a total of seven genes with evidence of transcription (S1B Table). In addition, evidence for protein expression was provided from both total proteomics analyses of trophozoites and pseudocysts (TvNlpC_A2 and TvNlpC_B4) [29] and from phagolysosome-enriched fractions (TvNlpC_A2 and TvNlpC_A3) [30] (S1C Table). TvNlpC_A2 was also detected in the secretome of *T. vaginalis* [31].

In an attempt to further resolve the origins of the TvNlpC genes, we investigated the existence of homologues in the draft genomes of *Trichomonas gallinae* [32] and *Trichomonas tenax* [33] and searched for additional similar bacterial homologues to the genes from the *Trichomonas* lineage. A total of eight and six NlpC/P60 homologues were identified in the draft genome of *T. gallinae* and *T. tenax*, respectively, and four NlpC/P60 homologues were also identified in the genome annotation of the trichomonad *Histomonas meleagridis* [34] (S1C Table). Phylogenetic analyses focusing on the *Trichomonas* spp and *Histomonas* NlpC/P60 protein sequences and a selection of their most similar bacterial homologues established that all three *Trichomonas* species have members of the subfamilies NlpC_A and NlpC_B and that these were acquired by a shared common ancestor through two likely distinct events of lateral gene transfer. Despite the generally poorly resolved topology, characterized by some

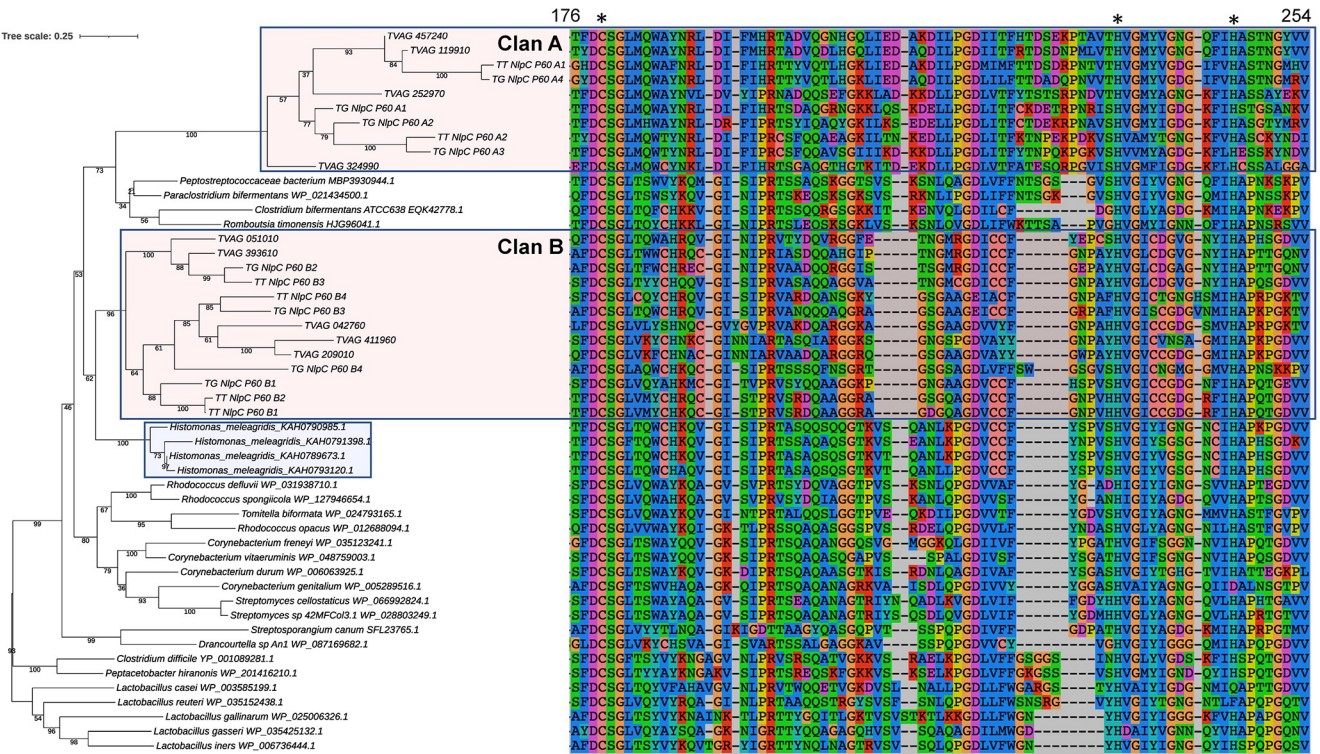

**Fig 1. Phylogenetic relationship of NlpC/P60 proteins from *Trichomonas*, *Histomonas* and a selection of their close bacterial homologues.** Phylogenetic relationship inferred by maximum likelihood from a protein alignment of 103 residues with the best fit model LG G4 I (chosen according to Bayesian Information Criterion) with 1000 ultrafast bootstrap replicates. Ultrafast bootstrap support values are indicated below the branches. The scale bar shows the inferred number of substitutions per site. Sequences from the *Trichomonas* species from Clan A and B are highlighted by the red boxes (TT, *Trichomonas tenax*; TG, *Trichomonas gallinae*; TVAG_XXXXXX, *Trichomonas vaginalis*). The blue box highlights the four homologues from *Histomonas meleagridis*. The accession numbers for the bacterial homologues are indicated following the species names. A segment of the protein alignment for these proteins is shown on the righthand side, which covers the catalytic sites. The numbers at the top of the alignment indicate the residues number for TVAG_457240. The position of the catalytic triad C-H-H are indicated by * at the top of the alignment.

branches with low bootstrap values that are unstable upon variation in the evolutionary models and taxa sampling used for phylogenetic inferences (as per two examples in S1 Fig), the phylogeny with a broader sampling of Trichomonads sequences (Fig 1) is consistent with and support our initial conclusions based on the *T. vaginalis* NlpC/P60 sequences only [21]. The lateral acquisition of NlpC/P60 genes from bacteria is a feature of *T. vaginalis* that is shared with other mucosal-dwelling Trichomonads.

## *T. vaginalis* NlpC/P60 genes participate on protozoa-lactobacilli interactions

The expansion of this gene family in *T. vaginalis*, and now revealed in other Trichomonads (Fig 1), suggests that these peptidoglycan hydrolases might be important for the interaction of these mucosal-dwelling protozoans with the local microbiota of the host. Here, we wanted to understand whether TvNlpC genes of clans A and B might share complementary activities on the interaction of *T. vaginalis* with host-protective vaginal lactobacilli. To test this, we incubated *T. vaginalis* alone or with vaginal *Lactobacillus gasseri* ATCC 9853 in a serum-free defined media (SFM). During the incubation period, samples were taken for microscopy, colony forming units (CFU) of lactobacilli and differential expression analysis of the nine TvNlpC genes (Fig 2).

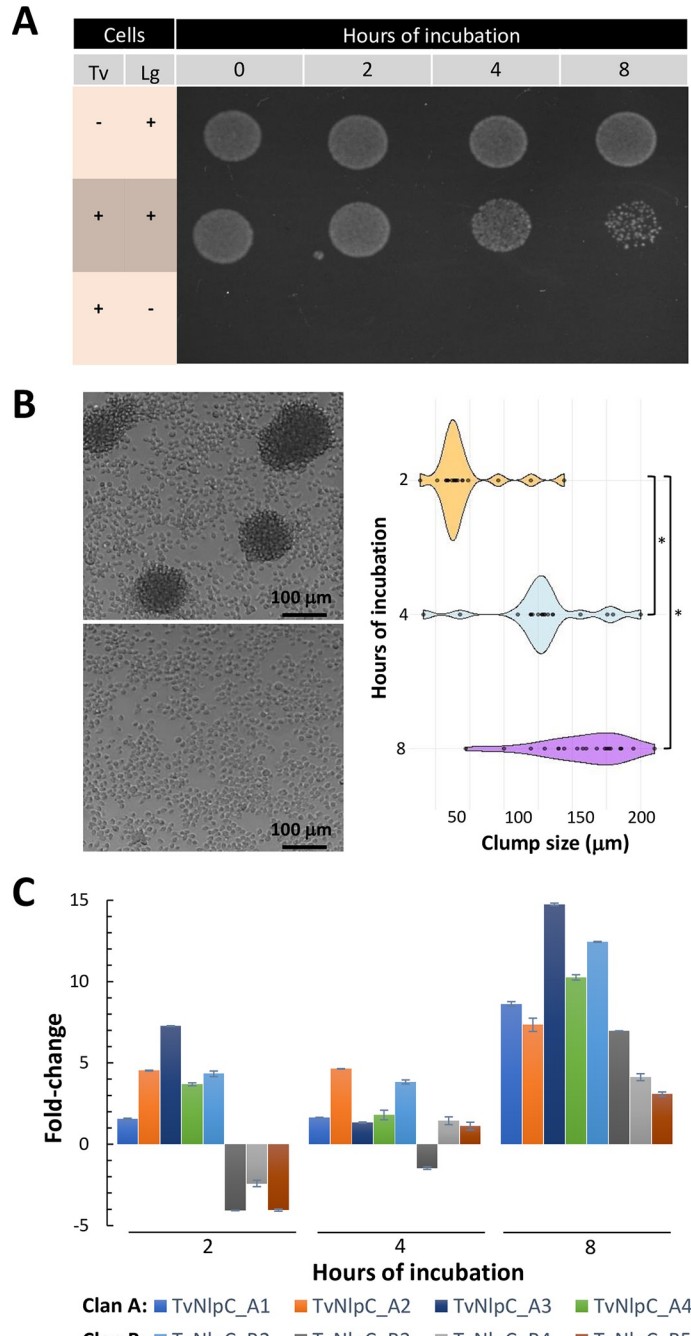

**Fig 2. The interaction of *Trichomonas vaginalis* (Tv) and *Lactobacillus gasseri* (Lg) results on the reduction of bacterial population, parasite aggregation and upregulation on the expression of TvNlpC/P60 genes. (A)** Spotting single and mixed cultures of Tv and Lg on MRS-agar plates (selective for lactobacilli) shows that the population of Lg is reduced in numbers when in mixed culture with the protozoan. **(B)** The interaction of Tv and Lg results in co-aggregation. Parasites clump together in the presence of the bacteria (top left), but not in the absence of them (bottom left), after 8 hours of incubation. The diameter of twenty random cell clumps show that their sizes vary and tend to increase with time of incubation (right, *p-value < 0.01 from Games-Howell pairwise test). **(C)** RT-qPCR assays show the fold-change on the expression levels of TvNlpC/P60 genes across time, when comparing Tv in the presence versus absence of Lg. All TvNlpC/P60 genes of clan A and B were included in this analysis, except for TvNlpC_B1 which did not produce a significant $C_T$ value. Results show that all genes with measurable expression are upregulated in *T. vaginalis* when in mixed culture with lactobacilli at 8 hours.

We noticed a reduction in the population of lactobacilli upon coincubation with *T. vaginalis*. This reduction in bacterial numbers was visibly different from the control (i.e. lactobacilli alone), particularly at 8 hours (Fig 2A). Notably, this reduction was accompanied by the formation of clumps of live parasites of varied sizes which increased with the time of incubation (Fig 2B). When comparing the expression of TvNlpC genes, despite some fluctuations, we found that all TvNlpC genes with detectable expression were upregulated when the protozoan was co-incubated with lactobacilli at the latest time point (Fig 2C). The maximum response was for TvNlpC_A3 with ~15-fold upregulation at 8 hours. The expression of TvNlpC_B1 was not detectable with a significant $C_T$ value and thus excluded. Together, these findings demonstrate that vaginal *L. gasseri* induced aggregation of parasites which concomitantly upregulate the expression of TvNlpC genes leading to a reduction in bacterial population.

We have previously shown a gain of function phenotype when overexpressing an unregulated copy of TvNlpC_A1 and A2 in *T. vaginalis*, with parasites gaining the ability of controlling populations of the model bacterium *Escherichia coli* in mixed culture [21]. Here, we expanded this analysis to TvNlpC_B3 and B5, two members belonging to the other gene family clan (S1 Table and Fig 1). To do this, all four transfected *T. vaginalis* overexpressing one of the four HA-tagged TvNlpC genes (A1, A2, B3 or B5) were co-incubated with lactobacilli (Fig 3). *T. vaginalis* transfected with an empty vector (i.e. not expressing any protein of interest) were used as control.

We observed that the unregulated expression of an additional copy of any of the TvNlpC genes in *T. vaginalis* enhanced the ability of the parasite to control the population of vaginal strain of *L. gasseri* in mixed cultures, with noticeable differences at 4 and 8 hours of incubation, as compared to the control (Fig 3A). We then undertook immunofluorescence, cell fractionation and Western blotting to localise the HA-tagged TvNlpC_B3 and _B5 in *T. vaginalis* cells (Fig 3B). TvNlpC_A1 and A2 were found on the surface and in the secretion of *T. vaginalis* [21, 31]. Here, TvNlpC_B3 and _B5 were detected as punctate spots in *T. vaginalis* cells by immunofluorescence. While staining for TvNlpC_B3 was uniformly dispersed in the cell, TvNlpC_B5 was found within larger areas by the nucleus (Fig 3B, top). This observation was consistent with the detection of both proteins mostly within the organellar fraction on Western blots (Fig 3B, bottom). Despite different cellular localizations, gain of function experiments indicate that TvNlpC genes of both clans A and B contribute to aiding the parasite to control populations of vaginal *L. gasseri* in mixed cultures.

## *T. vaginalis* NlpC_B3 and NlpC_B5 share substrate specificity

To investigate the substrate specificity of TvNlpC_B3 and TvNlpC_B5, purified recombinant proteins, and their predicted catalytically inactive versions TvNlpC_B3 (C53S) and TvNlpC_B5 (C52S), were incubated with PG isolated from two different bacterial strains. The *E. coli* MC1061Δ6LDT strain lacks all known LD-transpeptidases resulting on a PG that is rich in tetrapeptides [35]. The *E. coli* CS703-1 strain lacks multiple Penicillin-binding proteins resulting on a pentapeptide-rich PG [36]. A control sample contained no protein. The samples were then incubated with the muramidase cellosyl and the resulting muropeptides were analysed by HPLC, as described previously [37].

As expected, neither of the catalytically inactive versions were able to digest PG (Fig 4). Both TvNlpC_B3 and TvNlpC_B5 with native catalytic residue were active against both types of PG, showing the substantial reduction of the main substrate peaks. Both enzymes digested the cross-linked TetraTetra dimer but not the Tetra monomer in the tetrapeptide-rich PG (Fig 4A), producing the disaccharide dipeptide (Di, peak 2 in Fig 4) and the disaccharide tetrapeptide linked to D-Ala-m-Dap from a second peptide (Tetra- D-Ala-m-Dap, peak 1 in Fig 4).

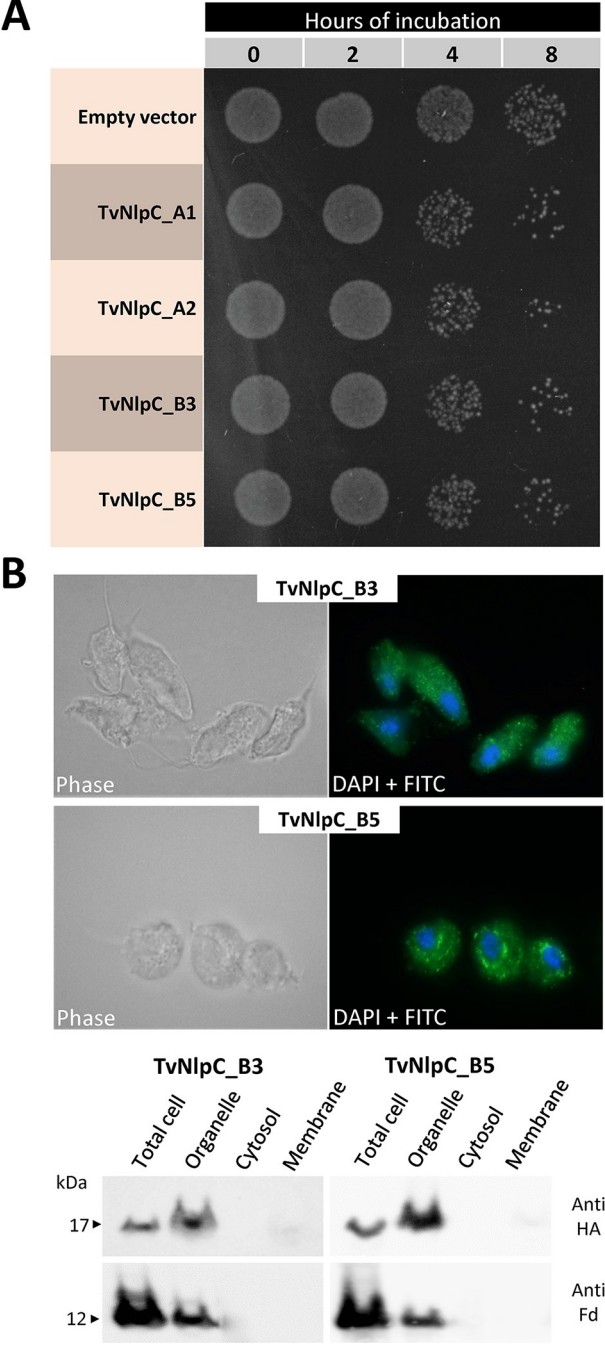

**Fig 3. Gain of function phenotype and cytolocalization of TvNlpC genes in *T. vaginalis*.** **(A)** Spotting mixed cultures of *T. vaginalis* and lactobacilli on MRS-agar plates reveals a greater reduction on bacterial numbers when the protozoan is expressing an unregulated, HA-tagged copy of TvNlpC_A1, TvNlpC_A2, TvNlpC_B3 or TvNlpC_B5 as compared to the empty vector. **(B)** Immunofluorescence microscopy and Western blot show that TvNlpC_B3 and _B5 are located in organelles. (Top panel) Immunofluorescence of *T. vaginalis* cells transfected with HA-tagged TvNlpC_B3 and _B5 and stained with FITC-conjugated antibody and DAPI. (Bottom panel) Cell fractionation of *T. vaginalis* transfected with HA-tagged TvNlpC_B3 and _B5 followed by Western blots, probed with antibodies against the HA-tagged protein (Anti-HA) or the organellar marker ferredoxin (Anti-Fd).

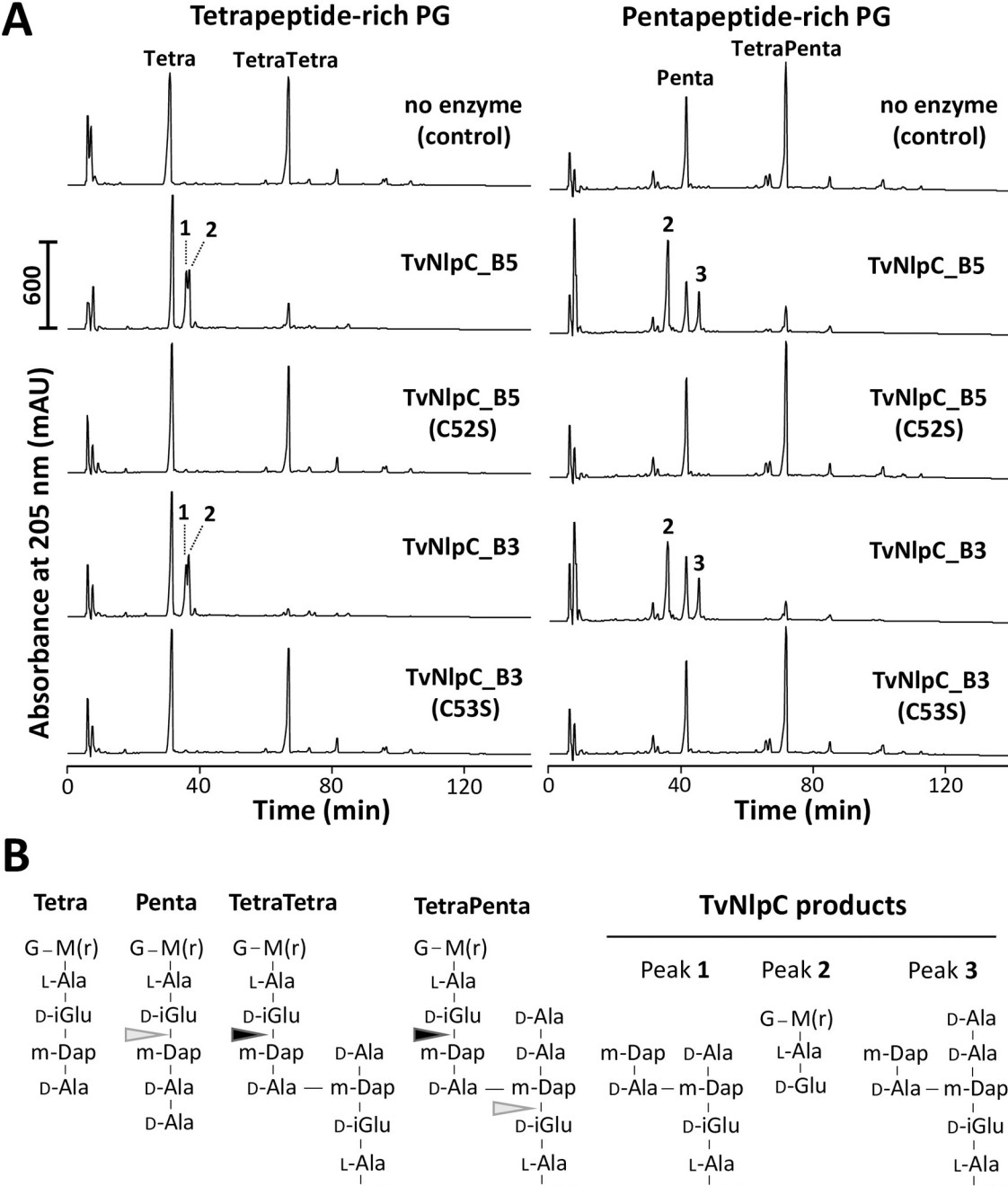

**Fig 4. Enzymatic activity of TvNlpC_B3 and TvNlpC_B5. (A)** The catalytic active and inactive versions of these enzymes, as indicated, were incubated with tetrapeptide-rich PG from *E. coli* MC1061Δ6LDT (left side) or pentapeptide-rich PG from strain CS703-1 (right side). Muropepides were released by cellosyl, reduced with sodium borohydride and separated by HPLC. Peaks of the main cleavage products were numbered 1 to 3. **(B)** Proposed structures of the main muropeptides and their TvNlpC products. Black arrows indicate sites of efficient cleavage detected from both TvNlpC enzymes. Grey arrows indicate less efficient cleavage sites. G, N-acetylglucsoamine; M(r), N-acetylmuramitol; L-Ala, L-alanine; D-iGlu, D-isoglutamate; D-Glu, D-glutamate; m-Dap, meso-diaminopimelic acid; D-Ala, D-alanine.

Both enzymes were also active on pentapeptide-rich PG, cleaving the TetraPenta dimer to produce Di and the disaccharide pentapeptide linked to D-Ala-m-Dap from a second peptide (peak 3 in Fig 4). In addition, both enzymes also cleaved in, albeit with less efficiency, monomeric or dimeric pentapeptide stems (light grey arrows in Fig 4B). When cleaving in the Penta monomer, the Penta-D-Ala-m-Dap product was detected (peak 3, Fig 4B). All products of cleavage were confirmed by mass spectrometry analysis. The *in vitro* endopeptidase activities of the recombinant TvNlpC_B3 and TvNlpC_B5 were also verified against purified PG of *L. gasseri* ATCC 9853 (S2 Fig).

These results confirm that both enzymes cleave between D-iGlu (position 2) and m-Dap (position 3) of pentapeptides or the donor part of cross-linked peptides, i.e. the same cleavage site as for Tse1, an 'attacking' PG hydrolase secreted by the type 6 secretion system of *Pseudomonas aeruginosa* [38, 39]. We have previously shown that TvNlpC_A1 and A2 were not capable of digesting pentapeptides in PG [21]. In conclusion, despite the redundant activities of TvNlpC_B3 and B5, they complement the activities of TvNlpC_A1 and A2.

## E64 binds to TvNlpC_B3 catalytic site *in vitro* and limits its activity in mixed cultures

We have previously shown that the NlpC/P60 domains of TvNlpC_A1 and _A2 adopt the expected papain-like fold [40] and exhibit open and highly accessible active sites [21]. In addition, the N-termini form two bacterial SH3 domains with a shared arrangement relative to the NlpC/P60 domains [21]. The SH3 domain is a feature lacking among TvNlpC proteins of the clan B [21], thus determining the structure of a TvNlpC of this clan could give us additional insights in the function of this gene family. Moreover, no structures of NlpC/P60 domain proteins have been determined with their ligand to date. Thus, the mechanisms of how these proteins achieve their specificity is unknown. To help elucidate key residues in the active site environment, we undertook crystallisation experiments with and without the cysteine protease inhibitor E64, obtaining crystals that were then used for structural characterization of the TvNlpC_B3 protein.

The final model of TvNlpC_B3, refined to an R/Rfree of 0.158/0.170 (full data collection and refinement statistics in S2 Table), and contains the contiguous segment from Asn23 to Tyr137 (Fig 5). As expected, TvNlpC_B3 exhibits an α+β, papain-like fold, consisting of 4 α-helices and 5 β-sheets arranged in antiparallel fashion. This fold is highly conserved across all NlpC/P60 and CHAP domains [41]. The active site residues of TvNlpC_B3 form a catalytic triad, identified as Cys53, His99 and His111 using homology with family members [22] [41], (Fig 1). Helix2 contains the active site cysteine at the N-terminus of the helix, positioning it in close proximity with His99 and His111 (Fig 5). His111 is also responsible for the single unusual rotamer of our model and is supported by well-defined electron density. Similarly to other Cys-His-His peptidases, the positioning of His111 appears to orientate His99 for facilitating enzymatic activity (via hydrogen bonding between ND1 atom of His111 and the NE2 atom of His99) [42].

The active site groove runs from the N-terminus of Helix α2, between β-strands 2 and 3 and across the β-sheet ending at strand 4 (Fig 5). This groove occupies a full face of the domain, making it a dominant feature of the otherwise globular protein (S3 Fig). The active site cysteine is located centrally within the grove, with the imidazole groups of histidine residues on one side, and the hydroxyl of Ser54 on the other, altogether forming the groove's floor. The serine side of the groove terminates on the R group of the highly conserved Arg71 (where the first SH3b domain of NlpC_A1 would interact with the NlpC/P60 domain) (Fig 5).

The structure of TvNlpC_B3 bound to the inhibitor E64 was determined by molecular replacement using the Apo enzyme structure, and refined to an R/Rfree of 0.183/0.198 (full

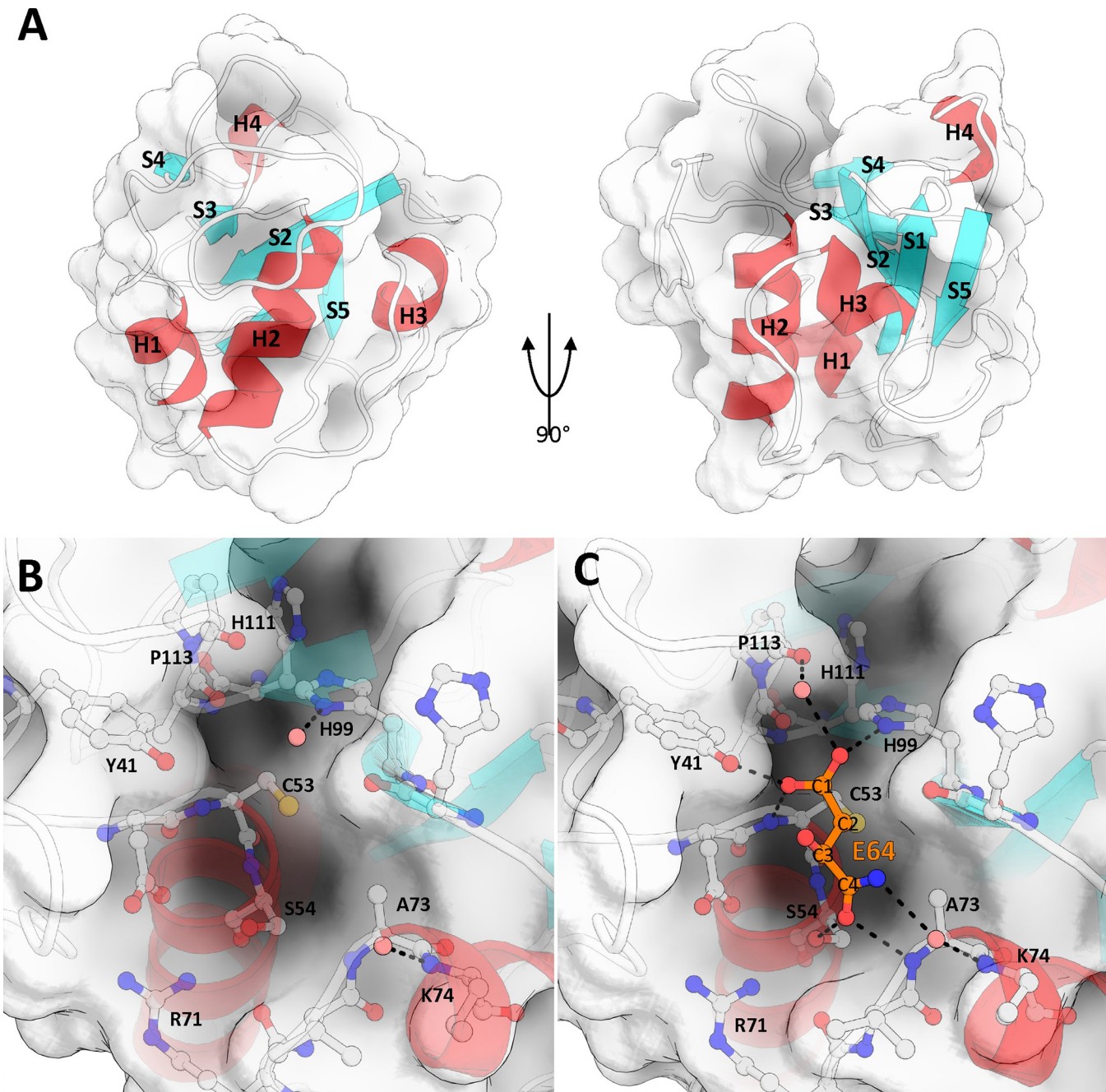

**Fig 5. The structure of TvNlpC_B3. (A)** Papain-like fold of TvNlpC_B3, consisting of 3 N-terminal helices, and an anti-parallel beta-sheet structure, shown here at 0 and 90 degrees. A single helical turn, H4, is present between S4 and S5. The fold forms a binding cleft centered at the N-terminal beginning of helix H2 and running between strands S2 and S3, terminating on S4. **(B)** Active site and nearby residues of TvNlpC_B3, Cys53, His99, and His111 form a catalytic triad. **(C)** Active site when bound by E64 fragment, E64 covalently binds to TvNlpC_B3 active site via C2 and the thiol of Cys53.

data collection and refinement statistics in S2 Table). Examination of the electron density in the active site revealed a small extra fragment of density connected to the active site cysteine (Cys53) sulfur atom (S4 Fig). While this fragment is considerably smaller that the entire E64 molecule, as explained ahead, we believe it is a fragment of the E64 molecule. Mass spectrometry of the sample used for crystallisation revealed masses for two major species which correlate

to TvNlpC_B3 bound to E64 (14,046 Da), and TvNlpC_B3 bound to a 131 kDa fragment (13,821 Da), with the latter being the major species (S4 Fig). The fragment of E64 present in our structure is well supported by density, and represents this 131 kDa fragment, suggesting that the breakdown of the E64 molecule and selective crystallization of this complex.

We modelled the E64 fragment covalently bound to the Cys53 via the C2 atom. The carboxyl group on C1 forms polar interactions with the carboxyl group of Tyr41, the ND1 atom of His99, and the main chain nitrogen of Cys53. The Tyr41 residue is conserved, and is suggested to have a role in regulating the nucleophilicity of Cys53, or in stabilising the substrate and/or tetrahedral intermediate during catalysis [43]. The carboxyl group from C3 of the E64 fragment forms no interactions with the protein and is exposed to solvent. The carboxyl from C4 forms a network of hydrogen bonds between sidechain of Ser54 and the mainchain nitrogen of Ala73. N1 is the last visible atom of the E64 fragment and co-ordinates a water molecule with the mainchain nitrogen of Lys74, this water would likely be displaced when considering the intact E64 molecule. No major differences in the protein were observed between the E64 bound, and apo TvNlpC_B3 structures RMSD of all atoms = 0.103) indicating a lack of conformational change on inhibitor binding.

The structure determination of TvNlpC_B3 with E64 (Fig 5) led us to evaluate whether this cell-permeable cysteine protease inhibitor might have a measurable effect on the ability of *T. vaginalis* to control populations of lactobacilli in mixed cultures. In addition, we wanted to see if the bactericidal activity of *T. vaginalis* observed in mixed cultures (Fig 2A) may apply to an important vaginal dysbiotic bacterium (i.e. *G. vaginalis*). To evaluate these hypotheses, co-incubation experiments of TvNlpC_B3-transfected *T. vaginalis* with vaginal *L. gasseri* or *G. vaginalis* were carried out in the absence or presence of E64 (Fig 6).

We confirmed that the unregulated expression of TvNlpC_B3 enhances the control of lactobacilli, but not of *G. vaginalis* (Fig 6A), by *T. vaginalis* in mixed cultures. E64 was seen to partially prevent the action of *T. vaginalis* on controlling lactobacilli populations. When quantifying these reductions on bacterial numbers by CFU (Fig 6B), we observed that *T. vaginalis* was up to ~8 times more effective on controlling lactobacilli when transfected with TvNlpC_B3 than with the empty vector. Although E64 did not completely block this action by the parasite, it inhibited the enhanced activity of TvNlpC_B3-transfected *T. vaginalis* against lactobacilli by up to ~9-fold at the latest time-point (Fig 6B). Therefore, in agreement with the ligand-protein structure, this cysteine protease inhibitor significantly reduces the ability of the protozoan on controlling populations of *L. gasseri* in mixed cultures.

## Discussion

In this study, we reported that the two-clan configuration of laterally acquired NlpC/P60-containing peptidases (i.e. NlpC_A and NlpC_B subfamilies) is also present in trichomonads other than *T. vaginalis*. The conservation of both subfamilies, with the preservation of the catalytic triad Cys-His-His, in the phylogeny of trichomonads with genomes available highlights two events of lateral gene transfer in a common ancestor that likely took place in a pigeon-infecting species [20, 44]. This was followed by gene duplications and amelioration, leading to optimized enzymes that may target various bacteria in the different context of hosts (i.e. pigeons or mammals) and mucosal surfaces (i.e. oral cavity and upper digestive tract or the urogenital tract). The structural configurations identified for the TvNlpC were also found among the sequences from *T. gallinae* and *T. tenax* with or without SH3b domains. In addition, the *H. meleagridis* was found to encode four homologues most similar to the Clan B enzymes, despite weak support values. *H. meleagridis* is an important bird pathogen that cannot be grown axenically, but requires Gram-negative bacteria to grow *in vitro* [34].

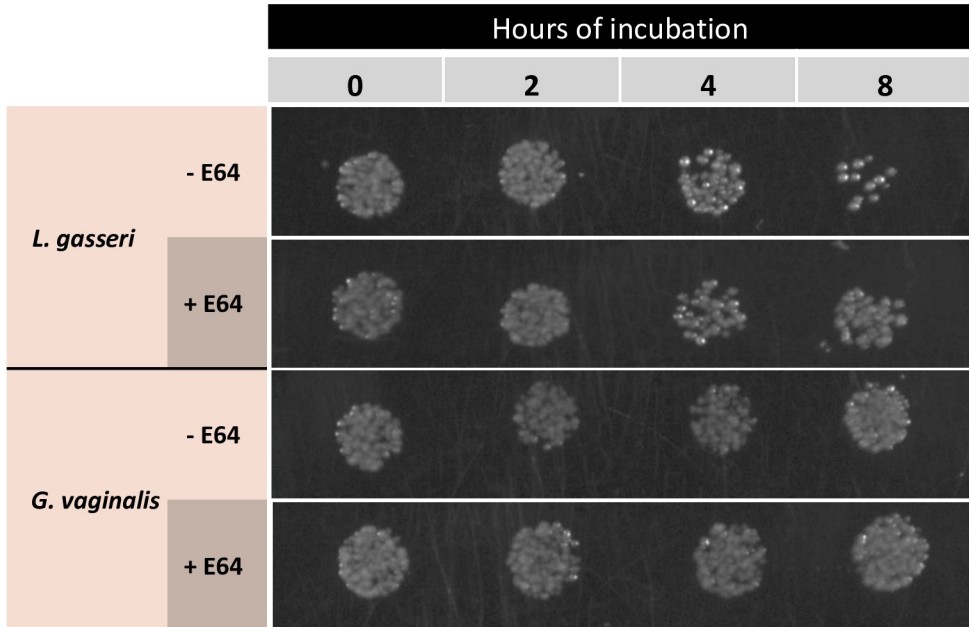

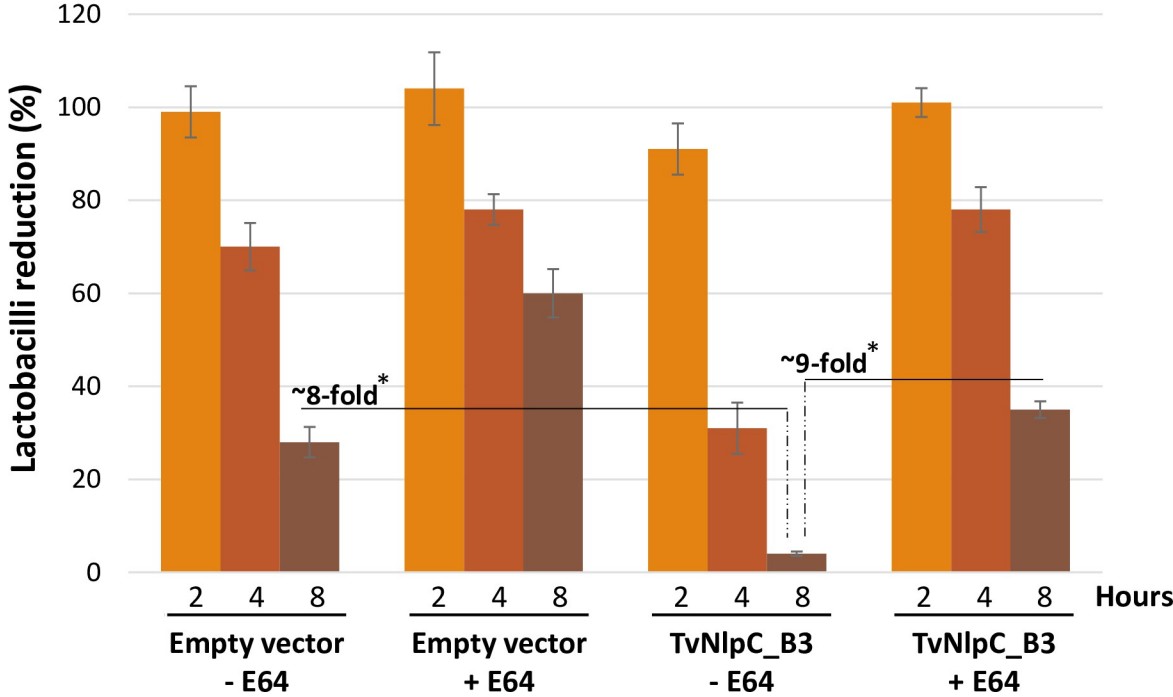

**Fig 6. The activity of TvNlpC_B3-transfected *T. vaginalis* against vaginal bacteria from mixed cultures.** (A) Spot plating reveals that TvNlpC_B3-transfected *T. vaginalis* reduces the population of *L. gasseri* but not of *G. vaginalis*, which reduction is partially reversed by E64. (B) CFU quantification from spread plates shows that the expression of TvNlpC_B3 enhances the reduction of lactobacilli population by up to ~8-fold as compared to the empty vector control, while E64 partially prevents the reduction of lactobacilli population by up to ~9-fold (*p-value < 0.01 from a t-test).

While this study showed that TvNlpCs might help *T. vaginalis* to compete out lactobacilli, in the case of *H. meleagridis*, these enzymes may be even more critical by aiding the parasite to feed on bacteria for some essential metabolites. Studies will be needed to assess the

function and the importance of NlpC/P60-containing peptidases among these trichomonads.

For *T. vaginalis*, we have previously shown that two NlpC/P60 enzymes of clan A (TvNplC_A1 and _A2) are functional DL-endopeptidases that cleave bacterial peptidoglycan (PG) with similar structure and redundant activity [21]. On the other hand, TvNlpCs of clan B lack SH3 domains. We depicted here the papain-like structure of the TvNlpC_B3 with a fragmented E64 molecule. To our knowledge, this is the first NlpC/P60-containing peptidase to be co-crystalized with a ligand shedding light in the structure-function of these enzymes. TvNlpC_B3 and _B5 were shown to digest tetra and pentapeptides in PG. TvNplC_A1 and _A2 were limited to digest tetrapeptides only, and thus possibly optimized to digest the mature form of PG on bacterial cell walls [21]. Hence, this study shows that the PG hydrolytic activities between TvNlpCs of clans A and B complement each other. More studies will be necessary for further understanding the specificity and complementary activities of all nine TvNlpC enzymes.

We now know as well that at least eight out of the nine TvNlpC genes are indeed expressed in *T. vaginalis* cells, complementing expression evidence that was initially based on transcriptomics and proteomics data generated from *T. vaginalis* in the absence of any bacteria [27–31]. Importantly, the expression of these genes respond positively to the presence of lactobacilli (with up to 15-fold upregulation) helping explain why previous omics studies may have missed on detecting the expression of some TvNlpC genes. For instance, TvNlpC_B3 and _B5 were never detected in previous proteomics studies [29–31]. The positive response of TvNlpC gene expression to lactobacilli indicates that *T. vaginalis* should recognize environmental cues, unknown at this stage, that might be related to this vaginal bacterium.

TvNlpC_B3 and _B5 were found to be located in organelles, whereas both TvNplC_A1 and _A2, were shown on the surface and to be secreted by *T. vaginalis* [21]. TvNlpC_B3 and _B5 were not found in the proteome of lysosome-enriched fractions [30], but lack of their detection in these organelles may be explained by not having bacteria triggering upregulation of TvNplC gene expression. For example, despite remarking similarity between the secreted TvNplC_A1 and _A2 [21], only the latter was detected in the *T. vaginalis* secretome [31]. Further studies will be needed for determining the final organellar destination of TvNlpC_B3 and _B5, as well as the cellular localization of other TvNlpCs.

*T. vaginalis* is known for being both secretory and phagocytic [45]. Phagocytosis by *T. vaginalis* has been reported elsewhere [46–49], involving the transitioning to the amoeboid morphology of this protozoan and other dramatic changes at ultrastructural cellular level [48, 49]. *T. vaginalis* has been shown to phagocyte host cells, yeast and bacteria to a different extent [46–49] and this broad phagocytic capacity was found to the correlate with the virulence of the strains [48, 49]. Limited evidence suggests that phagocytosis by *T. vaginalis* may contribute to pathogenesis, the vaginal biome changes and nutrition acquisition by the parasite. We envisage that TvNlpCs may be employed by this protozoan parasite to attack vaginal bacteria extracellularly or intracellularly following phagocytosis. However, more investigation is needed to understand how these extra- and intracellular TvNlpCs exactly reach the PG in the bacterial cell wall and the possible involvement of phagocytosis in this process.

We found that the endogenous upregulation of TvNlpC gene expression by lactobacilli is accompanied by a reduction of these bacteria in mixed cultures and that the overexpression of these enzymes in transfected cells enhances this phenotype. On the other hand, the unregulated expression of TvNlpC_B3 in *T. vaginalis* did not lead to reducing *G. vaginalis* population in mixed cultures. In alignment to the co-crystal structure of TvNlpC_B3 with E64, we showed that this cysteine-protease inhibitor supresses the action of *T. vaginalis* on lactobacilli significantly. This observation is in agreement with E64 reducing the cytotoxicity of *T. vaginalis* in

response to the secretion of cysteine-proteases [45]. This inhibitor, however, does not completely block the bacteriolytic action of *T. vaginalis* suggesting that additional factors other than TvNlpCs (e.g. amidases and antimicrobial peptides) might also contribute to the efficacy of this parasite controlling bacterial populations. Further studies are needed to understand the complementary activities of all nine NlpC/P60 enzymes of *T. vaginalis* and their preference for targeting vaginal bacteria with specificity.

Because PG is the major structural component of bacterial cell walls and essential for maintaining the integrity of bacterial cells, it is not surprising to find that employing PG-degrading enzymes (i.e. amidases and peptidases) for controlling bacterial population may be a widely spread strategy that has co-evolved with hosts across the tree of life and in concert with their autochthonous microorganisms. This strategy is often employed by bacteriophages [50], has been reported in predatory bacteria [51] and between non-predatory bacteria, with the latter delivering these enzymes via type IV and VI secretion systems [38, 52]. More recently, *Photobacterium damselae* subsp. *piscicida* (a pathogenic bacterium of marine fish) was shown to secrete PnpA, a NlpC/P60 with notable specificity [53]. PnpA was shown to cleave PG of competing bacteria, giving a competitive advantage to the pathogen. Differently to our work, instead of using purified PG, the authors employed insoluble sacculi which might help preserve the structural features of the cell wall required for targeting bacteria with specificity. PnpA is secreted and, similarly to TvNlpC_A1 and TvNlpC_A2 [21], it remains to be described how these enzymes reach the PG of their bacterial targets.

PG-degrading enzymes have been claimed as potential antibacterial genes that were laterally acquired by eukaryotes from bacteria [54]. These acquisitions have enhanced innate immune function, as described in the deer tick *Ixodes scapularis* that employs an amidase to limit the proliferation of the bacterial agent of the Lyme disease *Borrelia burgdorferi* [24]. In protozoan parasites, TvNplCs are the only examples of functional PG hydrolases described to date, and our study is showing that this might be a feature on the evolution of other Trichomonads infecting different mucosal sites and host species. Such investigations will help understand the complexity, dynamics and evolution of host-parasite-microbiome interactions. In perspective, further studies will help clarify how trichomoniasis is accompanied by vaginal dysbiosis which knowledge might contribute to the development of antimicrobial therapies that concomitantly reduce co-morbidities associated with this infection and the dysbiotic microbiome.

## Materials and methods

### Identification of NlpC/P60 genes in the draft genomes of *Trichomonas* spp and sequence comparisons

Local tblastn searches against the draft genome sequences of *T. tenax* (Strain NIH4 ATCC 30207, GenBank assembly accession: GCA_900231805.1) and *T. gallinae* (isolate XT1081-07, GenBank assembly accession: GCA_008369845.1) were performed with *T. vaginalis* NlpC/P60 proteins as query, identifying 6 and 8 NlpC/P60 ORFs for the respective species. ORFinder (https://www.ncbi.nlm.nih.gov/orffinder/) was used to identify full-length coding sequences of each gene candidate. BlastP searches at the NCBI were performed to identify close homologues among bacteria. These searches also identified hits derived from proteins annotated in the recently published genomes of *Histomonas meleagridis* [34]. The four proteins sequence from only one of two strains of *Histomonas meleagridis* were considered. Proteins aligned to homologues from *T. vaginalis* and selected proteins from bacteria using SEAVIEW [55] further supported the identification of full ORFs in *T. gallinae* and *T. tenax*. Slightly editing the original alignment (104 residues) [21] led to 103 well-aligned residues, i.e. matching hypothesis of site

homology across the selected bacteria and Trichomonads' sequences (S1 Data). Maximum likelihood phylogenetic inferences were performed with IQ-TREE and considered several evolutionary models, both homogenous models and protein mixture models [56, 57]. Protein domain organisation was analysed using InterProScan (https://www.ebi.ac.uk/interpro/search/sequence/) [58].

### Single and mixed cultures of protozoan and vaginal bacteria

*Trichomonas vaginalis* genome strain G3 was cultured in TYM medium [59] supplemented with 10% heat-inactivated horse serum, 10 U/ml penicillin, and 10 µg/ml streptomycin (Invitrogen). *Lactobacillus gasseri* ATCC 9857, a strain of human vaginal origin, was grown in MRS (deMan, Rogosa, and Sharpe) medium without serum. *Gardnerella vaginalis* ATCC 14018 was grown in 1685 NYC III medium supplemented with 5% of glucose and 10% heat-inactivated horse serum. All three microorganisms were grown at 37˚C without agitation in microaerophilic condition, i.e. in standard 15 ml Falcon tubes (12 cm high) filled up with 15.5 ml of media to reduce oxygen exchange. For the coincubation assays, the following procedures were taken. The number and viability of *T. vaginalis* cells, grown from frozen stocks in the absence of antibiotics for at least three days and with daily passages, were assessed under a hemocytometer. The number and viability of bacterial cells were assessed by flow cytometry as previously described [13]. Microbial media was removed by centrifugation and cells (>95% viability) were washed and resuspended in antibiotic- and serum-free keratinocyte media (K-SFM; Invitrogen). *T. vaginalis* ($2$–$5 \times 10^5$ cells/ml) was mixed with bacteria in a 1:3 ratio in 1 ml volume and in a 24-well tissue culture plate and incubated at 37 ˚C under anaerobiosis (Oxoid Anaero-Gen Compact, Thermo Scientific). As controls, bacteria and *T. vaginalis* were treated and incubated under the same conditions but in the absence of each other. Cultures were mixed by pipetting, before collecting samples for plating on agar. For the time points indicated, images of the microbes settled on the plastic surface of the wells were taken under the inverted microscope (EC3 Leica camera) at 200X magnification. Twenty random clumps of *T. vaginalis* were measured from their longest diameter distance. Additionally, for every time point, 5 µl of undiluted cultures were spotted and 50 µl of cultures diluted in sterile water were spread on MRS-agar or NYC-agar plates (which procedure is selective for lactobacilli or *G. vaginalis* respectively) and grown at 37˚C under atmospheric air for *L. gasseri* or under anaerobiosis for *G. vaginalis* (Oxoid AnaeroGen Compact, Thermo Scientific) for 24-48h. The spot plates were photographed, and CFU counts were obtained from the spread plates indicated, 0.1 mM E-64 was added to the mixed cultures at time zero from a 10 mM stock prepared in sterile water. Control wells without E-64 received the same volume of sterile water, instead. The introduction of an episomal copy of a HA-tagged TvNlpC_B3 and _B5 (TVAG_042760 and TVAG_411960, respectively) or an empty plasmid in *T. vaginalis* was achieved by transfection, as previously described [21]. Transfected *T. vaginalis* were used in the experiments above, where indicated. These co-incubation experiments were run three times independently, with two spread plating replicas for each experiment. Data was quantified across all experiments and results were expressed as a percentage of CFU reduction in comparison to the controls (i.e., in the absence of *T. vaginalis*).

### Gene expression analysis

For gene expression analysis, mixed cultures of protozoa:lactobacilli and single cultures of *T. vaginalis* were scaled up to 15 ml of K-SFM in one tissue culture dish for each time point. Total RNA was obtained from the mixed and single cultures at each time point by TRIzol (Invitrogen) and subsequently purified by removal of DNA (DNase I, Ambion) and clean-up

with RNeasy MinElute (Qiagen). We utilized the same gene nomenclature as previously published [21], i.e. TvNlpC_A1 (TVAG_119910), TvNlpC_A2 (TVAG_457240), TvNlpC_A3 (TVAG_324990) TvNlpC_A4 (TVAG_252970) for TvNlpC/P60 genes of clan A, and TvNlpC_B1 (TVAG_393610), TvNlpC_B2 (TVAG_051010), TvNlpC_B3 (TVAG_042760), TvNlpC_B4 (TVAG_209010) and TvNlpC_B5 (TVAG_411960) for TvNlpC/P60 genes of clan B. Reverse transcription and quantitative real-time PCR (RT-qPCR) were conducted using SuperScript IV Reverse Transcriptase and PowerUP SYBR green Master Mix (Thermo Fisher) in a 7900 HT-Realtime thermocycler (Applied Biosystems). We followed the PCR protocols as exactly detailed in our previous publication [21], with TvNlpC-specific primers listed in here (S3 Table), employing the standard threshold cycle ($C_T$) method for relative expression analysis after data normalization and using recommended software (SDS 2.3 and RQ Manager 1.2 applications, Applied Biosystems). The expression levels of the genes above were compared between mixed cultures versus single cultures from triplicate experiments.

## Subcellular localization of TvNlpC_B3 and _B5

Transfected *T. vaginalis* cells expressing the HA-tagged TvNlpC_B3 and _B5 were used to assess the subcellular localization of these proteins by two complementary assays. We employed indirect immunofluorescence assay (IFA) with the mouse anti-HA antibody and the FITC-conjugated secondary antibody and cell fractionation followed by Western blot, as described previously [21]. For the Western blots, protein from cell fractions (organelle, cytosol and membrane) and total cell were brought up to equivalent volumes. To help reinforce the lack of detection in any given fraction, loading volumes varied. When probing for the HA-tagged TvNlpC_B3 and _B5 with the anti-HA, we loaded 5X more volume of protein from cell fractions than total cell. When probing for ferredoxin (anti-Fd), a known organellar marker for *T. vaginalis*, this ratio was inverted. Other than this modification, protocols for IFA, cell fractionation and Western blot were followed as exactly detailed in our previous publication [21].

## Expression and purification of TvNlpC_B3 and _B5

Coding sequences for TvNlpC_B3 and _B5 were truncated at the N-terminus by 15 and 16 amino acids respectively, based on the identification of predicted signal sequences using software packages (Phobius and PrediSi) [60, 61]. Inactive enzyme mutants were generated for TvNlpC_B3 and TvNlpC_B5 by targeting the active site cysteine (C53S and C52S, respectively) identified via sequence homology with other family members. Sequences were cloned into pET47b expression plasmid with an N-terminal hexa His-tag and 3C-protease cleavage site (detailed in S2 Data), with proteins expressed in *Escherichia coli* strain BL21(DE3). Bacterial cells were grown in LB to an $OD_{600}$ of 0.6–0.8, and expression induced at 18˚C by the addition of 1 mM IPTG. Cells were harvested by centrifugation and resuspended in 20 mM TRIS/HCl pH 7.8, 300 mM NaCl, 0.5 mM TCEP, 10% v/v glycerol. Proteins were purified by immobilised metal affinity chromatography and eluted in 10 mM TRIS/HCl, pH 8.0, 300 mM NaCl, 300 mM Imidazole. To remove the N-terminal tag, pooled protein was incubated with 3C protease and dialysed into 10 mM TRIS/HCl pH 8.0, 150 mM NaCl, 0.1 mM TCEP overnight. Final purification was carried out by size exclusion chromatography using a Superdex 75 16/60 column. Fractions containing pure protein were pooled and concentrated using 3,000 molecular weight cut-off spin-concentrator and stored at -20˚C until required.

## Enzymatic assay for TvNlpC_B3 and _B5

PG from *Escherichia coli* strains MC1061Δ6LDT (tetrapeptide-rich PG) or CS703-1 (pentapeptide-rich PG) was incubated with different TvNlpC proteins (10 μM) in 20 mM Tris/HCl pH

7.5, 150 mM NaCl for 4 h at 37 ˚C in a thermomixer set at 800 rpm. A control sample contained no protein. The reaction was stopped by heating the samples at 100 ˚C for 10 min. To release the muropeptides from the PG, the pH was adjusted to 4.8 and muramidase cellosyl (Hoechst, Frankfurt am Main, Germany) was added to a final concentration of 50 μg/ml prior to incubation overnight at 37˚C in a thermomixer set at 800 rpm. The reaction was stopped by heating the samples at 100˚C for 10 min on a dry heat block and the sample was centrifuged at 13,000 ×g for 10 min. The supernatant was supplemented with an equal volume of 0.5 M sodium borate pH 9.0 and ~1 mg sodium borohydride powder, followed by an incubation of 30 min at ambient temperature to reduce the muropeptides. The samples were then acidified to pH 4.0–4.5 using 20% phosphoric acid. The resulting reduced muropeptides were separated by HPLC using a 250 × 4.6 mm 3 μm Prontosil 120-3-C18 AQ reversed-phase column (Bischoff, Leonberg, Germany) as described [37]. The eluted muropeptides were detected by their absorbance at 205 nm. Products of TvNlpC versions were collected and analysed by mass spectrometry as described [62, 63]. The measured neutral masses were: Di, 698.053 and 698.067 (theoretical: 698.286); Tetra-D-Ala-m-Dap, 1184.309 (theoretical: 1184.530); Penta-D-Ala-m-Dap, 1255.353 (theoretical: 1255.567). The PG of *L. gasseri* ATCC 9847 was purified as previously described for other Gram-positive bacteria [64], and subjected to the same treatments described above.

## Crystallisation and structure determination of TvNlpC_B3

Crystallisation of the purified recombinant TvNlpC_B3 was undertaken by sitting-drop vapour diffusion using the JSCG+, PACT Premier, and MORPHEUS screens (Molecular Dimensions) in Intelliplate HT 96 well plates with a 50 μL reservoir. Crystallisation drops consisting of a 1:1 and 2:1 protein:reservoir ratio in a total volume of 300 nL were dispensed using an Oryx automated robotic crystallisation system (Douglas Instruments). All crystallisation experiments took place at 18˚C. TvNlpC_B3 crystals were grown in JSCG+ condition E4; 1.26 M Ammonium sulfate, 0.1 M TRIS, pH 8.5. The cysteine protease inhibitors E64 was added to the purified TvNlpC_B3 in at least 5:1 molar ratio, and left for an hour to incubate at room temperature. The resulting mixture was then screened on the JCSG+ 96HT tray for crystal growth. Crystals grew in the same conditions as the apo crystals and were cryo-protected with the addition of 20% w/v glycerol, before flash-frozen and stored under liquid nitrogen to be shipped to the Australian Synchrotron (Melbourne, Victoria, Australia) for data collection using the MX1 and MX2 beamlines. Data were indexed and integrated using either MosFlm software suite [65] or XDS software package. Integrated reflections were used as input to the AIMLESS pipeline of the CCP4 software suite [66, 67]. A 5% Free-R reflection set was maintained for TvNlpC_B3 models with and without E64.

Crystallographic phases for TvNlpC_B3 were estimated by molecular replacement with PHASER [68]. A truncated NlpC/P60 domain of TvNlpC_A1 (PDBid: 6BIO) [21] was used as a search model. Molecular replacements were chosen by a TFZ value over 8.0, and RFZ over 4.0. Successful molecular replacement was visualised in COOT software [69], and the electron density maps scrutinised for interpretable electron density with logical differences between the residues of the search model, and the expected residues of the crystallised protein. Structural models were refined with iterative cycles of refinement in REFMAC5 [70] from the CCP4 software suite. After REFMAC refinement, the model and associated density, as well as Fo-Fc density maps were visualised for real-space refinement and manipulation in COOT software [69]. Refinement cycles were monitored with crystallographic R/Rfree factors, and RMS bond angle and bond length. Due to the ratio of unique observations to atom parameters in the final model (observ/param = 2.47) and high resolution of the dataset, full anisotropic B factors were

included in the final refinement. The final model was validated using the Molprobity web server (http://molprobity.biochem.duke.edu/), and the PDB submission validation web server (https://validate-rcsb-1.wwpdb.org/).

## Supporting information

**S1 Fig. A comparison of phylogenetic relationship between bacterial and Trichomonads NlpC/P60 protein sequences from two distinct evolutionary models.** Different models and taxa sampling were used to investigate the phylogenetic relationships between the identified NlpC/P60 protein sequences from the three species of *Trichomonas*: *T. vaginalis*, *T. tenax* and *T. gallinae*. Sequences from the fourth Trichomonad *Histomonas meleagridis* were also included in addition to close bacterial homologues to the Trichomonads homologues. Maximum likelihood based phylogenetic inferences used: (i) the best fitting identified homogenous model for the protein alignment using the automatic model selection function as implemented in iq-tree (see Material and Methods sections for mode details) (model LG+G+I, shown on the right and that also corresponds to the phylogeny shown in Fig 1) or (ii) protein mixture models, one such phylogeny is shown on the left (LG4X+F) that allow different amino acid composition and evolutionary rates across the alignment. The protein mixture models were used, as they are considered to be more reliable in extracting phylogenetic signal from divergent sequences. The branches with the lower support values (from 1000 ultrafast bootstraps) are also those corresponding to sections of the phylogenies that are sensitive to the model used. These differences between these two phylogenies highlight the lack of phylogenetic signal in that short alignment of only 103 residues.
(TIF)

**S2 Fig. Endopeptidase activity of TvNlpC_B3 and TvNlpC_B5 against purified peptidoglycan of *L. gasseri*.** The catalytic-active and -inactive versions of TvNlpC_B3 (A) and TvNlpC_B5 (B), as indicated, were incubated with PG from *L. gasseri*. Muropeptides were released by cellosyl, reduced with sodium borohydride and separated by HPLC. Cleavage products (red arrows), that are absent from the -inactive enzyme controls, indicate activity of these enzymes against the PG of *L. gasseri*.
(TIF)

**S3 Fig. A comparison of the active site groove between TvNlpC_A1 and TvNlpC_B3.** This comparison shows that TvNlpC_B3 exhibits a more open and accessible groove shape than TvNlpC_A1, including an absence of the nearby extended loop structure between strands S1 and S2 (highlighted blue). A nearby SH3 domain (highlighted pink) modulates the groove in TvNlpC_A1.
(TIF)

**S4 Fig. Electron density of the TvNlpC_B3 active site before and after refinement with fragment of E64.** For reference, 2Fo-Fc is shown at 1.0 sigma blue, positive and negative Fo-Fc at 3.0 sigma are shown in green and red respectively. (A) Apo TvNlpC_B3, (B) TvNlpC_B3 with E64 prior to and after building of E64 fragment, and (C) final model of TvNlpC_B3 with E64 fragment modelled. (D) Mass spectroscopy analysis of TvNlpC_B3 incubated with E64. Three major peaks were observed corresponding to TvNlpC_B3 (peak 1), TvNlpC_B3 and the E64 fragment (peak 2) and TvNlpC_B3 and E64 (peak 3). (E) The fragment identified in both the crystallography and mass spectroscopy analysis corresponds to a 131Da fragment of E64.
(TIF)

**S1 Table. Features of NlpC/P60 genes and proteins of Trichomonads with draft genomes available.** (A) Features of the nine TvNlpC genes and proteins of clans A (A1 to A4 in blue) and B (B1 to B5 in red); (B) Evidence for the expression of TvNlpCs at transcript and protein level as previously reported; (C) NlpC/P60 genes identified in Trichomonads: *Trichomonas gallinae*, *Trichomonas tenax* and *Histomonas meleagridis*.
(XLSX)

**S2 Table. Data collection and refinement statistics for the crystal structure of TvNlpC_B3 with and without the inhibitor E64.**
(DOCX)

**S3 Table. List of forward and reverse primers for the nine TvNlpC genes, which were used in quantitative real-time PCR.**
(XLSX)

**S1 Data. Data supporting site homology hypothesis across the selected bacteria and Trichomonads' sequences used for phylogeny.**
(FST)

**S2 Data. TvNlpC_B3 and TvNlpS_B5 amino acid sequences detailing the N-terminal signal truncations used for *E. coli* cloning, expression and purification of these recombinant proteins.**
(DOCX)

## Author Contributions

**Conceptualization:** Waldemar Vollmer, Robert P. Hirt, Augusto Simoes-Barbosa, David C. Goldstone.

**Data curation:** Michael J. Barnett, Jeremy R. Keown, Jacob Biboy, Joe Gray, Waldemar Vollmer, Robert P. Hirt, Augusto Simoes-Barbosa.

**Formal analysis:** Michael J. Barnett, Jully Pinheiro, Jeremy R. Keown, Jacob Biboy, Ioana-Wilhelmina Lucinescu, Waldemar Vollmer, Robert P. Hirt, Augusto Simoes-Barbosa, David C. Goldstone.

**Funding acquisition:** Waldemar Vollmer, Augusto Simoes-Barbosa, David C. Goldstone.

**Investigation:** Michael J. Barnett, Jully Pinheiro, Jacob Biboy, Ioana-Wilhelmina Lucinescu, Augusto Simoes-Barbosa.

**Methodology:** Michael J. Barnett, Jully Pinheiro, Jeremy R. Keown, Jacob Biboy, Joe Gray, Ioana-Wilhelmina Lucinescu, Augusto Simoes-Barbosa.

**Project administration:** Waldemar Vollmer, Augusto Simoes-Barbosa, David C. Goldstone.

**Resources:** Waldemar Vollmer, Robert P. Hirt, Augusto Simoes-Barbosa, David C. Goldstone.

**Supervision:** Waldemar Vollmer, Robert P. Hirt, Augusto Simoes-Barbosa, David C. Goldstone.

**Validation:** Michael J. Barnett.

**Visualization:** Michael J. Barnett, Jacob Biboy, Robert P. Hirt.

**Writing – original draft:** Michael J. Barnett, Waldemar Vollmer, Robert P. Hirt, Augusto Simoes-Barbosa, David C. Goldstone.

**Writing – review & editing:** Michael J. Barnett, Jully Pinheiro, Jeremy R. Keown, Jacob Biboy, Ioana-Wilhelmina Lucinescu, Waldemar Vollmer, Robert P. Hirt, Augusto Simoes-Barbosa, David C. Goldstone.

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
