## [Decision Letter · Decision Letter 0]

14 Feb 2023

Dear Dr. Simoes-Barbosa,

Thank you very much for submitting your manuscript "NlpC/P60 peptidoglycan hydrolases of Trichomonas vaginalis have complementary activities that empower the protozoan to control host-protective lactobacilli" for consideration at PLOS Pathogens. As with all papers reviewed by the journal, your manuscript was reviewed by members of the editorial board and by several independent reviewers. In light of the reviews (below this email), we would like to invite the resubmission of a significantly-revised version that takes into account the reviewers' comments.

The additional data should be included in the revision: (1) include data showing that the NlpC/P60 hydrolases produced by T. vaginalis break down the peptidoglycan of Lactobacillus, (2) compare the killing effect of T. vaginalis on several bacteria, not just lactobacilli, to rule out a generalized bactericidal activity of the parasite, (3) data addressing whether phagocytosis is responsible for lactobacillus killing and (4) report T. vaginalis viability at the endpoint of experiments where the parasite is incubated at high concentrations for long periods.

All minor issues raised by the reviewers should also be addressed.

We cannot make any decision about publication until we have seen the revised manuscript and your response to the reviewers' comments. Your revised manuscript is also likely to be sent to reviewers for further evaluation.

Sincerely,

Patricia J. Johnson, PhD

Academic Editor

PLOS Pathogens

Dominique Soldati-Favre

Section Editor

PLOS Pathogens

Kasturi Haldar

Editor-in-Chief

PLOS Pathogens

orcid.org/0000-0001-5065-158X

Michael Malim

Editor-in-Chief

PLOS Pathogens

orcid.org/0000-0002-7699-2064

The additional data should be included in the revision: (1) include data showing that the NlpC/P60 hydrolases produced by T. vaginalis break down the peptidoglycan of Lactobacillus, (2) compare the killing effect of T. vaginalis on several bacteria, not just lactobacilli, to rule out a generalized bactericidal activity of the parasite, (3) data addressing whether phagocytosis is responsible for lactobacillus killing and (4) report T. vaginalis viability at the endpoint of experiments where the parasite is incubated at high concentrations for long periods.

All minor issues raised by the reviewers should also be addressed.

Reviewer's Responses to Questions

**Part I - Summary**

Reviewer #1: The authors describe the role of dipeptidoglycan/hydrolases produced by the protozoan Trichomonas vaginalis, and hypothesize the role of these enzymes in vaginal dysbiosis, specifically describing the effect of these enzymes on lactobacilli. In addition, they describe the active site of the enzymes.

The paper is clear, well written, all experimental procedures are clearly described, and the conclusions are very interesting.

I have only to report some minor points, which might improve the understanding of the paper for a reader unfamiliar with the mechanisms of pathogenicity of T.vaginalis.

Reviewer #2: Previous work has demonstrated that the vaginal microbiome is altered and dysbiotic in patients infected with the protozoan T. vaginalis, but those studies have not yet shown the mechanism underlying this connection. Here, the authors suggest a potentially causal relationship between T. vaginalis infection and the change in vaginal microbiome composition. Building upon their own previous studies, the authors show that T. vaginalis has the capability to directly antagonize bacteria through production of bacterial peptidoglycan-targeting NlpC/P60 hydrolases. Furthermore, they show that this organism, as well as several evolutionarily related organisms, produce two separate classes of NlpC/P60 hydrolases that break down peptidoglycan in different ways. The authors show a) that expression of T. vaginalis NlpC/P60 hydrolases is upregulated in the presence of lactobacilli and b) that survival of lactobacilli is decreased by co-culture with T. vaginalis (and this bacterial inhibition is enhanced when the T. vaginalis overexpresses NlpC/P60 hydrolases). Together, these results suggest that T. vaginalis may exert influence on the vaginal microbiome by producing enzymes that create an unfavorable environment for lactobacilli. I am pleased recommend publication after the authors address some key issues listed below.

**Part II – Major Issues: Key Experiments Required for Acceptance**

Reviewer #1: No Major Issues

Reviewer #2: The authors do not provide, or speculate about, a mechanism through which T. vaginalis inhibits lactobacilli specifically. One might guess that the NlpC/P60 hydrolases produced by T. vaginalis directly break down the peptidoglycan of Lactobacillus species; however, the authors do not show this. Instead, the authors show that the T. vaginalis NlpC/P60 hydrolases can break down the peptidoglycan of two genetically engineered strains of E. coli. The work could be greatly improved by performing the experiments shown in Figure 4 with peptidoglycan isolated from Lactobacillus (and ideally other known bacterial members of the vaginal microbiome) rather than just that of E. coli.

The authors show that lactobacilli are killed in the presence of T. vaginalis, but do not show the effects of T. vaginalis on any other bacteria. Without these key comparisons, the results in Figure 2A could be due to a general bactericidal activity of T. vaginalis.

While gain of function phenotype was demonstrated by overexpressing one of the four TvNlpC_A1, A2, B3 and B5, the loss of function was not demonstrated. Would be good if the authors could generate and evaluate TvNlpC deletion strains, if possible.

**Part III – Minor Issues: Editorial and Data Presentation Modifications**

Reviewer #1: Results:

The experiments described demonstrate a reduction of lactobacilli in the supernatant. Is it possible that the number of bacteria is reduced as a result of phagocytosis by protozoa? This possibility needs to be discussed, along with the possibility of phagocytosis activation under the experimental conditions. Do the authors have data that can describe whether phagocytosis-related genes are upregulated under the experimental conditions described?

Lane 206-277. This section is very long. Is it possible to reduce the number of words?

Discussion:

The role of phagocytosis of lactobacilli in reducing the number of these bacteria needs to be better discussed

Material and Methods:

Lane 383. The authors state that the culture conditions are microaerophilic. They should report at least the height of the tubes, to demonstrate the distance of the cultured cells from the interface of air and oxygen

Lane 389: The number of cells is very high: the concentration of 2-5 x 106, especially for long incubation periods, is not physiological for T.vaginalis. The authors should report cell viability under these conditions

396. Is the sample collected after shaking the cultures? If the cultures are not shaken, the bacteria tend to sediment.

Reviewer #2: Figure 2B. Quantification of cell clumps size does not provide much of additional value as there is no statistical comparison and there is a variability in sizes.

Figure 3B. Scale bars are missing.

Line 102: Authors claim that crystal structure was obtained with bound ‘biological ligand’. E64 inhibitor is broad spectrum cysteine peptidase inhibitor (papain, cathepsin B, cathepsin L, calpain and staphopain) isolated from Aspergillus japonicus. E64 is not a natural ligand for TvNlpCs.

Figure 3. WB. Anti-Fd. More details are required to describe what is Anti-Fd (from material and method section it is ferredoxin).

Line 182. TvNlpC_B3 and B5 could act on E.coli PG. Why not Lactobacillus PG? If TvNlpC_B3 and B5 can cleave Lactobacillus PG, it will provide insights on mechanism of controlling Lactobacillus population.

Figure 4. Typos for labels: B3 and B3 (C53S) mislabeled as B5 and B(C52S).

Line 233. Typo: ‘S2 Fig’ instead of ’Fig. S2’

Line 249. Though LC-MS data of TvNlpC_B3 with bound E64 inhibitor was mentioned, no data was demonstrated to confirm covalent biding of E64 to TvNlpC_B3. Moreover, a good negative control would be LC-MS data of TvNlpC_B3(C53S) + E64.

PLOS authors have the option to publish the peer review history of their article (what does this mean?). If published, this will include your full peer review and any attached files.

Reviewer #1: **Yes: **Pier Luigi Fiori

Reviewer #2: No
---

## [Decision Letter · Decision Letter 1]

18 Jul 2023

Dear Dr. Simoes-Barbosa,

We are pleased to inform you that your manuscript 'NlpC/P60 peptidoglycan hydrolases of Trichomonas vaginalis have complementary activities that empower the protozoan to control host-protective lactobacilli' has been provisionally accepted for publication in PLOS Pathogens.

Best regards,

Patricia J. Johnson, PhD

Academic Editor

PLOS Pathogens

Dominique Soldati-Favre

Section Editor

PLOS Pathogens

Kasturi Haldar

Editor-in-Chief

PLOS Pathogens

orcid.org/0000-0001-5065-158X

Michael Malim

Editor-in-Chief

PLOS Pathogens

orcid.org/0000-0002-7699-2064

Reviewer Comments (if any, and for reference):

Reviewer's Responses to Questions

**Part I - Summary**

Reviewer #1: The new version of the manuscript is definitely improved. The authors have responded to criticism

Reviewer #2: (No Response)

**Part II – Major Issues: Key Experiments Required for Acceptance**

Reviewer #1: The manuscript may be published in the present form

Reviewer #2: While the authors have not addressed all the Reviewers' critiques experimentally, I appreciate the new data, revisions and discussion that are not included. I am pleased to recommend acceptance of the manuscript.

**Part III – Minor Issues: Editorial and Data Presentation Modifications**

Reviewer #1: The manuscript may be published in the present form

Reviewer #2: (No Response)

PLOS authors have the option to publish the peer review history of their article (what does this mean?). If published, this will include your full peer review and any attached files.

Reviewer #1: **Yes: **Pier Luigi Fiori

Reviewer #2: No

---

## [Editor Report · Acceptance letter]

8 Aug 2023

Dear Dr. Simoes-Barbosa,

We are delighted to inform you that your manuscript, "NlpC/P60 peptidoglycan hydrolases of *Trichomonas vaginalis* have complementary activities that empower the protozoan to control host-protective lactobacilli," has been formally accepted for publication in PLOS Pathogens.

Best regards,

Kasturi Haldar

Editor-in-Chief

PLOS Pathogens

orcid.org/0000-0001-5065-158X

Michael Malim

Editor-in-Chief

PLOS Pathogens

orcid.org/0000-0002-7699-2064